# ARF3-Mediated Regulation of *SPL* in Early Anther Morphogenesis: Maintaining Precise Spatial Distribution and Expression Level

**DOI:** 10.3390/ijms241411740

**Published:** 2023-07-21

**Authors:** Qi Yang, Jianzheng Wang, Shiting Zhang, Yuyuan Zhan, Jingting Shen, Fang Chang

**Affiliations:** State Key Laboratory of Genetic Engineering, Ministry of Education Key Laboratory of Biodiversity Sciences and Ecological Engineering and Institute of Biodiversity Sciences, Institute of Plant Biology, School of Life Sciences, Fudan University, Shanghai 200438, China

**Keywords:** anther morphogenesis, *ARF3*, *SPL*, microsporogenesis, transcriptional regulation, *Arabidopsis thaliana*

## Abstract

Early anther morphogenesis is a crucial process for male fertility in plants, governed by the transcription factor SPL. While the involvement of AGAMOUS (AG) in *SPL* activation and microsporogenesis initiation is well established, our understanding of the mechanisms governing the spatial distribution and precise expression of *SPL* during anther cell fate determination remains limited. Here, we present novel findings on the abnormal phenotypes of two previously unreported *SPL* mutants, *spl-4* and *spl-5*, during anther morphogenesis. Through comprehensive analysis, we identified ARF3 as a key upstream regulator of *SPL*. Our cytological experiments demonstrated that ARF3 plays a critical role in restricting *SPL* expression specifically in microsporocytes. Moreover, we revealed that ARF3 directly binds to two specific auxin response elements on the *SPL* promoter, effectively suppressing AG-mediated activation of *SPL*. Notably, the *arf3* loss-of-function mutant exhibits phenotypic similarities to the *SPL* overexpression mutant (*spl-5*), characterized by defective adaxial anther lobes. Transcriptomic analysis revealed differential expression of the genes involved in the morphogenesis pathway in both *arf3* and *spl* mutants, with ARF3 and SPL exhibited opposing regulatory effects on this pathway. Taken together, our study unveils the precise role of ARF3 in restricting the spatial expression and preventing aberrant SPL levels during early anther morphogenesis, thereby ensuring the fidelity of the critical developmental process in plants.

## 1. Introduction

The anther is a vital male reproductive organ in plants responsible for the production of male gametophytes. Proper development of anthers is crucial for successful plant sexual reproduction and ensuring crop yield [1,2]. Anther morphology in angiosperms is highly conserved, characterized by four lobes and a well-defined development process [3,4,5,6,7]. In *Arabidopsis thaliana*, the stamen primordium initially forms as a spherical structure composed of three layers of somatic cells, with archesporial cells giving rise to primary sporogenous cells (PSC) internally and primary parietal cells (PPC) externally. Subsequently, PPCs differentiate into secondary parietal cells (SPC), and the SPCs further differentiate into the middle layer (ML), endothecium (En), and tapetum (T), while sporogenous cells (SSC) become microsporocytes [1,2]. The establishment of adaxial and adaxial polarity, as well as the correct differentiation of early cells, are crucial for anther morphogenesis.

Significant progress has been made in understanding anther morphogenesis and discovering key genes and regulatory mechanisms involved [8,9,10,11,12]. The MADS-BOX transcription factor SPL is considered a core regulator during early anther morphogenesis. The loss-of-function mutant *spl*-*1*, obtained from the *Ac/Ds* mutant library, exhibits anthers composed of highly vacuolated parenchyma cells, leading to the absence of anther lobes and the microsporocytes, resulting in male sterility [13]. mRNA in situ hybridization experiments have demonstrated specific expression of *SPL* in developing microsporocytes [13]. AG directly binds to the CArG-box element downstream of the *SPL* coding region, activating *SPL* and thereby initiating microsporogenesis. The *ag*-*1* mutant fails to form microspores, while *SPL* can induce microspore production in the absence of *AG*, indicating the importance of *SPL* in the microsporocytes [14]. *SPL* also plays a significant role in anther organogenesis. Ectopic expression of *SPL* leads to the transformation of petals into stamens in the weak mutant of the *APETALA2* (*AP2*), *ap2*-*1*. Ectopic expression of *SPL* overexpression also affects the expression of *AG*, *SEP3*, and *AP2*, suggesting that *SPL* is involved in the regulation of stamen organ identity determination [15]. *BAM1* and *BAM2* encode two functionally redundant receptor-like kinases that are essential for normal cell division and differentiation during early anther development. The *bam1 bam2* mutant lacks endothecium, middle layer, and tapetum and exhibits morphological characteristics of microsporocytes within the epidermis, indicating the involvement of *BAM1* and *BAM2* in cell fate determination. Further research has revealed a negative feedback regulation between *BAM1/2* and SPL, with SPL promoting microsporocytes preferential expression of *BAM1/2* and BAM1/2 restricting *SPL* expression to the microsporocytes [16]. *SPL* is not only involved in the division and differentiation of archesporial cells but also directly participates in early anther polarity establishment. 

*MPK3* and *MPK6* encode two functionally redundant protein kinases. The *mpk3/+mpk6/*-mutants exhibited a deficiency in forming the two adaxial anther lobes. Immunofluorescence analysis revealed the co-expression of MPK3, MPK6, and SPL in the microsporocytes. Subsequent investigations demonstrated the direct interaction between MPK3, MPK6, and SPL, leading to the phosphorylation of SPL. This phosphorylation event plays a regulatory role in the early establishment of polarity during anther development [17]. *PHB*, encoding a transcription factor, specifically governs the adaxial development of lateral organs in *Arabidopsis*. Studies have shown that *microRNA165/6* negatively regulates the expression of *PHB*. In situ hybridization results revealed opposite expression patterns between *microRNA165/6* and *PHB* during early anther development while exhibiting similar expression patterns with *SPL*. Further investigations have demonstrated that PHB inhibits *SPL* expression by directly binding to the *SPL* promoter, thereby influencing the formation of anther dehiscence zones [8]. In addition, *microRNA165/6* also regulates the expression of *UCL1* to control anther dehiscence [18]. 

*SPL* also exhibits responsiveness to various hormonal processes within anthers. BZR1, a key transcription factor in the brassinosteroid signaling pathway, and its homologous gene mutant, exhibit similar defective anther lobe phenotypes to the *spl* mutant. RT-PCR and in situ hybridization experiments indicated that *BZR1* and its homologue transcription factors regulate early anther morphogenesis through its effects on *SPL* [19]. The expression of the auxin transport protein gene, *PIN1*, is reduced in *spl* mutant [20]. In addition, *spl*-*D*, an endogenous overexpression mutant of *SPL*, shows a leaf polarity defective phenotype with severe leaf curling. Further investigations found significantly downregulated accumulation of auxin and expression of auxin synthesis genes, *YUCCA2* and *YUCCA6*, in *spl*-*D* [21].

The establishment of adaxial–abaxial polarity identity, a crucial aspect of proper anther morphogenesis, is regulated by two classes of antagonistic genes [22]. *ARF3*, also known as *ETTIN*, is essential for abaxial polarity establishment [3,10,11,23]. ARF3 physically interacts with an abaxial regulator, *ABERRANT TESTA SHAPE* (*ATS*), which encodes a KANADI (KAN) transcription factor. This interaction forms a complex that mediates the specification of abaxial cell fate [24]. In rice, an adaxial identity gene *OsPHB3*, encoding an HD-ZIP III family transcription factor, is expressed in the adaxial domain where stomium differentiation occurs, while OsETT1 is expressed in the abaxial domain formed between thecae [3]. *ARF3* and its homologous gene *ARF4* redundantly contribute to organ asymmetry by modulating KANADI activity [23]. Both *ARF3* and *ARF4* are suppressed by a class of *ta*-*siRNAs* (*tasiR*-*ARF*) derived from *TAS3* [25,26,27]. *TasiR*-*ARFs* move from the adaxial interface to the abaxial interface, creating a gradient of small RNAs that patterns the abaxial expression of *ARF3* in leaf primordia [28,29]. Through ChIP-seq and RNA-seq data analysis, 663 putative direct targets of ARF3 have been found [30]. However, the direct target of ARF3 during early anther development remains unknown.

Taken together, SPL is crucial for anther morphogenesis and AG has been established as a positive regulator of SPL during the initial stages of stamen initiation, while the negative regulator determinants acting upstream of SPL remain elusive. In this investigation, we identified ARF3 as a negative regulatory factor governing *SPL* expression based on a series of comprehensive analysis. ARF3 exerts its regulatory effects by directly interacting with the *SPL* promoter, thereby restricting the expression of *SPL* in the microsporocytes, curbing excessive and deleterious *SPL* expression levels, and safeguarding the accurate progression of early anther morphogenesis.

## 2. Results

### 2.1. The Critical Role of SPL Expression Level in Anther Morphogenesis

Previous investigations revealed that *SPL* is required for early sporogenesis and germ cell initiation [31]. In this study, we identified and characterized two novel T-DNA insertion mutants of *SPL* (Figure 1A), namely *spl*-*4* and *spl*-*5*. RT-PCR results showed that *spl*-*4* represents a knockout mutant, while *spl*-*5* exhibited an overexpressed phenotype (Figure 1B). To assess the impact of *SPL* knockout and overexpression on anther morphogenesis and the establishment of four anther lobes, we examined the morphology of mature anthers in these mutants. Scanning electron microscopy results showed that the mature anthers of the wild-type exhibited the typical four-lobe structure (Figure 1C), with mature fertile pollen within the sporangia (Figure 1F). In contrast, *spl*-*4* displayed significant abnormalities in anther polarity establishment (Figure 1D), resulting in the absence of pollen and complete infertility (Figure 1G). Notably, *spl*-*5* displayed dysplasia phenotype in the two adaxial anther lobes, leading to abnormal anther polarity (Figure 1E), although fertile pollen remained unaffected (Figure 1H). These findings underscore the fact that both *SPL* functional deficiency and overexpression can cause varying degrees of abnormal anther morphogenesis, suggesting the critical importance of maintaining the appropriate expression level of *SPL*. 

To gain further insights into the role of *SPL* during anther morphogenesis, we conducted semi-thin section experiments to examine the early morphology of *spl*-*4* anthers. In the wild-type anthers, the primary parietal cells and primary sporogenous cells are formed at stage 3 (Figure 1I,R). Subsequently, at stage 5–6, the primary parietal cells differentiate into four layers of somatic cells, while the secondary sporogenous cells develop into microsporocytes (Figure 1J,K,S,T). In comparison to the previously reported phenotype of the *SPL* loss-of-function mutant *spl*-1 with no lobe formed, we observed two distinct phenotypes in *spl*-*4*. Type I *spl*-*4* anthers exhibited evident vacuolization and nuclear shrinkage in the primary parietal cells and primary sporogenous cells at stage 3 (Figure 1L,U). At stages 5 to 6, these anthers displayed pronounced vacuolization and failed to form anther lobes (Figure 1M,N,V,W), similar to the *spl*-*1* phenotype. Intriguingly, type II *spl*-*4* anthers exhibited no apparent abnormalities in the primary parietal cells and primary sporogenous cells at stage 3 (Figure 1O). At stage 5 to 6, these anthers formed a single anther lobe consisting of four layers of somatic cell cells and one layer of germ cells (Figure 1P,Q).

### 2.2. ARF3 Suppresses SPL Expression in Arabidopsis Anthers

Previous studies demonstrated that AG positively regulates *SPL*, thereby initiating microsporogenesis [14]. Given the crucial requirement of *SPL* for maintaining the correct expression, as deviations in either direction lead to an abnormal anther polarity development, it becomes imperative to identify negative regulatory factors acting upstream of *SPL*. We first explored the upregulation of *SPL* expression resulting from gene mutations using available published data. Transcriptome analysis revealed that elevated *SPL* expression in *arf3* mutant, and ChIP data showed that ARF3 can bind to the *SPL* promoter [30,32]. To validate the regulatory role of ARF3 in *SPL* expression, we assessed the expression level of *SPL* in the *arf3*-*29* mutant. The results unequivocally demonstrated significant up-regulation of *SPL* expression in *arf3*-*29* compared to the wild-type (Figure 2A). To further ascertain the direct regulation of *SPL* by ARF3, we detected *SPL* expression after treatment with dexamethasone (DEX) in *ARF3*::*ARF3*-*GR arf3*-*29* plants. DEX treatment induces the translocation of ARF3-GR fusion protein into the nucleus, thereby restoring the phenotype of *arf3*-*29* [33]. RT-PCR results showed a slight downregulation of *SPL* expression after 4 h of DEX treatment (Figure 2B), with a statistically significant downregulation observed after 6 h of DEX treatment (Figure 2B), providing compelling evidence for the direct regulatory role of ARF3 in moduling *SPL* expression. 

To investigate the spatial regulation of *SPL* by ARF3, we examined the expression pattern of *SPL* in wild-type and *arf3*-*29* mutants. In the wild-type, *SPL* exhibited preferential expression in the archesporial cells, secondary sporogenous cells, and microsporocytes at stage 2 to 5 (Figure 2C–E). However, in *arf3*-*29*, the expression of *SPL* was observed to be preferentially located to one side of the archesporial cells at stage 2 (Figure 2G). At stage 4, *SPL* expression was concentrated in the secondary parietal cells, secondary sporogenous cells, and the middle region surrounding the adaxial and abaxial lobes on one side (Figure 2H). By stage 5, *SPL* expression was detected in the microsporocytes, middle layer, the endothecium, and the tapetum (Figure 2I), indicating an expanded expression range of *SPL* in *arf3*-*29*. Subsequently, we examined the expression pattern of *ARF3* in early anthers. mRNA in situ hybridization revealed preferential expression of *ARF3* in the vascular tissues during stage 2 to 5 (Figure 2K–M), with slight expression observed in the microsporocytes at stage 5 (Figure 2M). These findings indicated an inverse expression pattern between ARF3 and SPL in anthers. The aforementioned results together suggested that *ARF3* exerts negative regulation on *SPL* at both the transcriptional and spatial levels.

### 2.3. ARF3 Is Required for the Establishment of Four Anther Lobes

Anther morphogenesis occurs from the Shoot Apex stage to Flower Stage 9. Analysis of publicly available data revealed high expression levels of *ARF3* and *SPL* during this period (Figure 3A), suggesting their crucial roles in the process of anther morphogenesis. To further investigate the function of *ARF3* in anther development, we examined the anther polar developmental phenotype of *arf3*-*29* mutants. Compared to wild-type anthers that display the characteristic four-lobed structure (Figure 3B), the *arf3*-*29* mutants showed defects in anther lobe formation (Figure 3C,D). Cytological experiments further confirmed the presence of anther lobe defects in *arf3*-*29* mutants (Figure 3E–G), highlighting the involvement of ARF3 in regulating anther morphogenesis. 

Given the phenotypic similarity between *ARF3* loss-of-function mutants and *SPL* overexpression mutants, we aimed to investigate whether *ARF3* and *SPL* are part of the same signaling pathway. To address this, we generated the *arf3*-*29 spl*-*1* double mutant through hybridization. Scanning electron microscopy analysis revealed similar phenotypes between *spl*-*1* and *arf3*-*29 spl*-*1*, characterized by significant abnormalities in polarity development (Figure 4A–C) and an inability to form pollen (Figure 4D–F). Furthermore, semi-thin section results demonstrated highly vacuolated anther cells and the absence of any anther lobes in both mutants (Figure 4G–I). These results strongly suggest that *ARF3* acts upstream of *SPL* in the regulation of the pathway governing anther development.

### 2.4. ARF3 Directly Interacts with the SPL Promoter and Suppresses AG-Mediated SPL Activation 

To investigate the direct interaction between ARF3 and *SPL* promoter, we performed chromatin immunoprecipitation (ChIP) assays in vivo. Floral primordia of *ARF3*::*ARF3*-*GFP 35S*::*AP1*-*GR cal ap1* inflorescences were treated with DEX for 5 days to reach floral stage 9 (anther stage 5). The results demonstrated that ARF3 is preferentially bound to the P1 promoter elements rather than P2, with higher binding observed at anther stage 5 (Figure 5A,B). To further confirm the direct binding of ARF3 to the *SPL* promoter, we performed electrophoretic mobility shift assays (EMSA). As shown in Figure 5C, ARF3 directly bound to the P1 elements of the *SPL* promoter, and both TGTCTC boxes were found to be essential for the interaction, while ARF3 did not bind to elements P2 of *SPL* promoter (Figure 5D). A previous study reported that AG directly activates *SPL* during microsporogenesis [14]. Therefore, we investigated whether ARF3 affects the transcriptional activation of AG on the *SPL* promoter using a transient transcription assay. In this assay, we connected 1 kb 3′-untranslated region to a 3.8 kb promoter of *SPL* (Figure 5E). The expression level of the *SPL* promoter-LUC construct was approximately 8-fold higher in its presence compared to the control (GFP). However, when ARF3 and AG were co-expressed, ARF3 repressed this increase in expression (Figure 5F). 

### 2.5. ARF3 and SPL Oppositely Regulate Early Anther Morphogenesis

To investigate the functional role of ARF3 and SPL in anther morphogenesis, we conducted a transcriptome analysis. In the *arf3*-*29* mutants, 1312 (73% of a total of 1807 differentially expressed genes) were found to be down-regulated, while 495 (27%) were up-regulated compared to the wild-type. Similarly, in the *spl*-*1* mutants, the expressions of 3479 genes were down-regulated, and 3999 genes were up-regulated (47% and 53% of the 7478 total altered genes, respectively) (Figure 6A). Among these, 739 genes were identified as commonly differentially expressed between *arf3*-*29* and *spl*-*1* (Figure 6B), with 28.7% exhibiting opposite regulatory patterns (11.6% of genes are up-regulated in *arf3*-*29* and down-regulated in *spl*-*1*, while 17.1% of genes are down-regulated in *arf3*-*29* and up-regulated in *spl*-*1*) (Figure 6C–E). Gene Ontology analysis revealed that the common DEGs which regulate in opposite directions (*arf3*-*29* down *spl*-*1* up and *arf3*-*29* up *spl*-*1* down) are enriched in morphogenesis related pathways (Figure 6F). These findings indicate that *ARF3* and *SPL* exert opposing regulatory effects on anther morphogenesis. Interestingly, the genes specifically regulated in *arf3*-*29* or *spl*-*1* were also enriched in morphogenesis-related pathways (Figure 6G), indicating that *ARF3* and *SPL* each contributed to the regulation of anther morphogenesis through distinct pathways. 

To further validate the opposing regulation of anther development by ARF3 and SPL, we examined the expression of marker genes for anther development. *bHLH010*, *bHLH089*, *bHLH091*, *AMS*, *MS1*, *MYB35* serve as markers for anther development, while *ROXY2* and *ROXY3* are associated with early anther morphogenesis. In *arf3*-*29* mutants, these genes were up-regulated, whereas in *spl*-*1* mutants, they were down-regulated, supporting the opposite regulatory roles of *ARF3* and *SPL* in anther development (Figure 7A–H).

## 3. Discussion

The process of organogenesis in plants differs from that in animals. While animal organs are typically formed during embryogenesis, plants continuously undergo organogenesis from stem cells throughout their life cycle [34,35,36,37]. Early morphogenesis, which involves the development of plant organs from their initial primordia into polar structures, is a crucial stage. Different from leaf development, which requires once polar establishment, anther early morphogenesis requires once polar establishment and once polar reversal [38], and ultimately forms a four-lobe structure. Therefore, anther development represents an excellent model to study plant organogenesis and morphogenesis, as it requires both polar establishment and polar reversal to form a four-lobed structure consisting of somatic and germ cells. 

In this study, we investigated the role of *SPL* in early anther morphogenesis. We identified two mutants, *spl*-*4* and *spl*-*5*, with disrupted *SPL* expression (Figure 1A–C). The *spl*-*4* anthers exhibited abnormal anther polarity dysplasia and no pollen production, while the *spl*-*5* had significantly reduced size in two adaxial lobes (Figure 1E,F,H). These results indicated that functional deficiency or overexpression of *SPL* can lead to a different degree of abnormal anther polarity development. Through analysis of the public database, we identified ARF3 as an upstream regulatory element of *SPL*. We found that ARF3 negatively regulates the expression of *SPL* at both the transcriptional and spatial levels. In the absence of *ARF3*, *SPL* expression was up-regulated, and its expression domain expanded beyond the microsporocytes to include the tapetum, middle layer, and endodermis (Figure 2A–J). Moreover, the lack of *ARF3* function resulted in a defect in the number of anther lobes (Figure 3B–G). By generating the *arf3*-*29 spl*-*1* double mutant (Figure 4A–I), we confirmed that *ARF3* acts upstream of *SPL* in the same signaling pathway. Next, we confirmed that ARF3 can directly bind to the *SPL* promoter in vivo. 

Through chromatin immunoprecipitation (ChIP) assays and electrophoretic mobility shift assays (EMSA) (Figure 5). we demonstrated that ARF3 directly binds to the P1 region of the SPL promoter, specifically to the TGTCTC elements. We also found that ARF3 suppresses the activation effect of AG (AGAMOUS) on *SPL* in a transient transcription assay. Transcriptome analysis of *arf3*-*29* and *spl*-*1* mutants revealed differentially expressed genes, particularly those with opposite expression patterns, which were significantly enriched in morphogenesis-related pathways. This suggests that ARF3 and *SPL* regulate anther morphogenesis in opposite directions and contribute to anther morphogenesis through distinct pathways. Previous studies highlighted the positive regulatory roles of AG and BES1 (BRI1-EMS-SUPPRESSOR 1) on SPL during microsporogenesis; negative regulatory factors upstream of SPL were poorly understood. Our identification of ARF3 as a negative regulator provides new insights into the regulatory network of SPL in anther polarity development.

In conclusion, this study enhances the understanding of the regulatory mechanisms involved in early anther morphogenesis. The ARF3-SPL module plays a crucial role in anther polarity development, but additional pathways are also involved. The research sheds light on the complex processes underlying plant organogenesis and provides new evidence and insights into the regulatory network governing early anther morphogenesis.

## 4. Materials and Methods

### 4.1. Plant Materials and Growth Conditions

*Arabidopsis thaliana* plants of L*er* or Col-0 background were planted in a greenhouse under conditions of a 16 h light/8 h dark cycle at 22 °C and 65% humidity. All mutants and transgenic *A*. *thaliana* lines are in the L*er* or Col-0 background. The *arf3*-*29* [39] *ARF3*::*ARF3*-*GR arf3*-*29* [33] and *spl*-*1* [13] were previously described. The *spl*-*4* (SAIL_827_A10) and *spl*-*5* (SALK_044645) mutants were obtained from the Nottingham Arabidopsis Stock Centre. The *arf3*-*29 spl*-*1* double mutants were generated by crossing the related single mutants. 

### 4.2. Phenotypic Analysis of Anthers

To assess pollen viability, stage 12 anthers were collected and subjected to staining with Alexander solution at 65 °C for 1 h [40]. Images of the stained anthers were captured using a digital camera (Nikon, Kyoto, Japan). For the transverse semi-thin sections assay, inflorescences were embedded in Spurr resin, cut into 1 μm thick slices, stained with toluidine blue, and visualized using a Nikon digital camera. Scanning electron microscopy was performed to photograph stage 12 anthers using a TM-3000 (Hitachi, Kyoto, Japan) scanning electron microscope. 

### 4.3. mRNA In Situ Hybridization

Gene-specific RNA probes were synthesized by in vitro transcription using the DIG RNA Labeling Kit (Roche, Basel, Switzerland). The samples were fixed in FAA (3.7% formaldehyde, 5% acetic acid, 50% ethanol) and embedded in wax. The embedded inflorescences were sectioned into 7 μm slices and subjected to dewaxing, rehydration, and dehydration. The sections were then hybridized with the prepared RNA probes at 55 °C overnight. After washing in SSC buffer, the slices were incubated with an anti-digoxigenin-AP antibody (Roche, Basel, Switzerland) at room temperature for 2.5 h. Hybridization signals were detected using NBT/BCIP color reaction (Roche).

### 4.4. ChIP-qPCR Analysis

Chromatin was isolated from 40 g of frozen tissue and fragmented by sonication to yield approximately 500 bp fragments. Immunoprecipitation was performed by incubating the chromatin with anti-GFP beads (Chromotek, Munich, Germany) overnight at 4 °C. The protein–chromatin complexes were then washed four times to eliminate nonspecific binding. The precipitated DNA was quantified by qRT-PCR. Primers specific for ChIP-qPCR assays are provided in Appendix A. The binding level was determined by calculating the ratio between IP and MOCK samples, normalized to the internal control ACT.

### 4.5. Electrophoretic Mobility Shift Assay

The coding regions of ARF3 were cloned into the pGEX4T-1 vector, and the resulting GST fusion protein was expressed in *E*. *coli* Rosetta cells. The empty pGEX4T-1 vector was used as a control. Biotin-labeled and unlabeled DNA probes containing the auxin responsive elements (TGTCTC) in the *SPL* promoter were designed. In vitro binding experiments were performed using the Light Shift Chemiluminescent EMSA system (Thermo Scientific, Waltham, MA, USA). The reaction mixtures were incubated in binding buffer (10 mM Tris-HCl (pH 7.5), 40 mM KCl, 2.5 mM MgCl_2_, 1 mM EDTA, 3 mM DTT, and 10% (*v*/*v*) glycerol) on ice for 30 min. The samples were then electrophoresed on 6% native polyacrylamide gel at 4 °C. The primers used for the EMSA probes are listed in Appendix A.

### 4.6. Transient Transcription Dual-Luciferase Assay

The coding regions of ARF3 and AG were cloned into p1306-GFP binary vectors. The promoter sequence and 3′-end non-coding sequence of *SPL* were connected and cloned into pGreenII-0800-LUC vectors. Agrobacterial cell suspensions containing TF protein(s) and a promoter construct were mixed at a ratio of 4:1 and infiltrated into young leaves of *Nicotiana benthamiana* using a published method [41]. After 48 h of cultivation, the leaf cells were lysed with Passive lysis buffer, and the luciferase activity was measured using the Dual-Luciferase assay kit (Promega, Madison, WI, USA). The firefly luciferase luminescence was detected, followed by the measurement of after quenching. The primers used for all constructs are listed in Appendix A. 

### 4.7. RNA Sequencing and Data Analysis

Approximately 20 inflorescences containing stage 1–9 anthers from L*er* and *arf3*-*29* plants were collected and immediately frozen in liquid nitrogen. Total RNA from each sample was extracted using the ZR plant RNA Miniprep™ kit (Zymo Research, Irvine, CA, USA) [30]. Then 3 μg total RNA of each sample was subjected to deep sequencing using an Illumina™ Hi-seq 2000 system (Illumina Ins., San Diego, CA, USA). The sequencing data were mapped and analyzed following a previously reported method [30]. The transcriptomic data of *spl*-*1* were obtained from a published study [42].

### 4.8. Quantitative Real-Time PCR Analysis

*Arabidopsis* inflorescences from various mutants were frozen in liquid nitrogen. RNA was extracted using the TRIzol (Invitrogen, Carlsbad, CA, USA) method and treated with DNase I (Takara, Kyoto, Japan). The cDNA was obtained with a reverse transcription system (Takara). Real-time PCR was performed using SYBR premix Ex Taq II (Takara) on a Bio-Rad Real-Time system (Bio-Rad, Hercules, CA, USA), with primers listed in Appendix A.

## Figures and Tables

**Figure 1 ijms-24-11740-f001:**
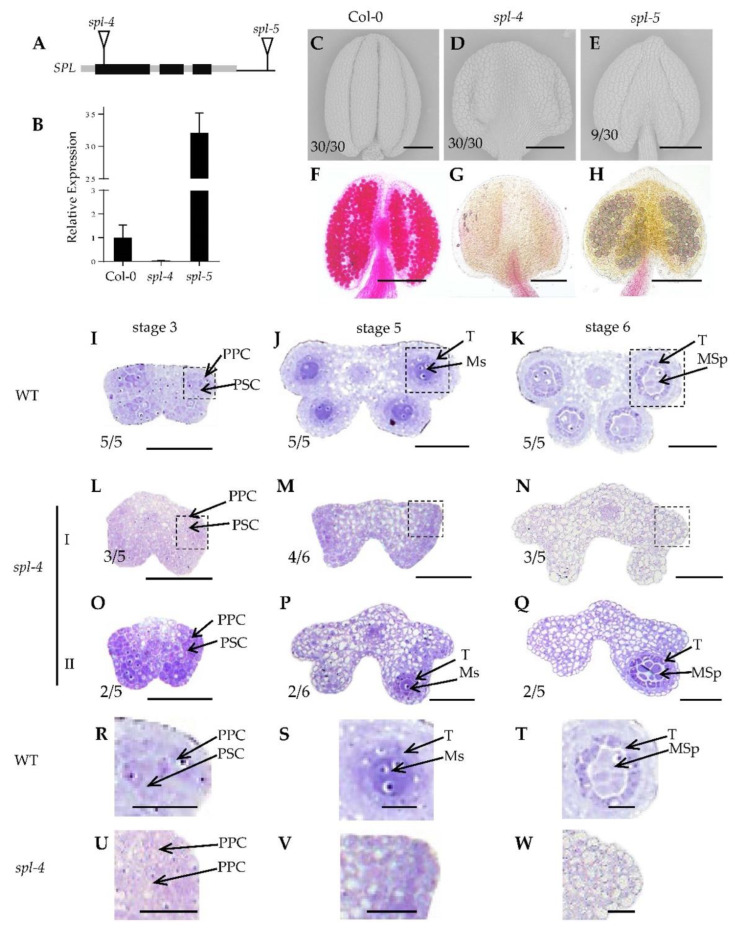
Phenotypic analysis of anthers from the wild-type, *spl-4*, and *spl-5* mutants. (**A**) Description of *spl-4* and *spl-5* mutants. Black boxes represent exons; grey boxes represent introns and untranslated regions, respectively. (**B**) Real-time PCR analysis showing the relative expression of *SPL* in *spl-4* and *spl-5*. (**C**–**E**) Scanning electron microscopy images of Col-0 (**E**), *spl-4* (**D**), and *spl-5* (**E**) anthers. (**F**–**H**) Alexander red staining of pollen grains from Col-0 (**F**), *spl-4* (**G**), and *spl-5* (**H**). (**I**–**Q**) Observation of anther morphology of wild-type (**I**–**K**) and *spl-4* (**L**–**Q**) using semi-thin transverse sections. I shows the type I *spl-4* anthers that produce no anther lobe, and II indicates the type II *spl-4* anthers that contain one anther lobe. (**R**–**T**) A closer view of the specific areas highlighted in (**I**–**K**). (**U**–**W**) A closer view of the specific areas highlighted in (**L**–**N**). PSC, primary sporogenous cells; PPC, primary parietal cells; Ms, microsporocytes; T, tapetum; MSp, microspores. Bars = 100 μm (**C**–**H**), 50 μm (**I**–**Q**), 15 μm (**R**–**W**). The number represents the proportion of the corresponding phenotype.

**Figure 2 ijms-24-11740-f002:**
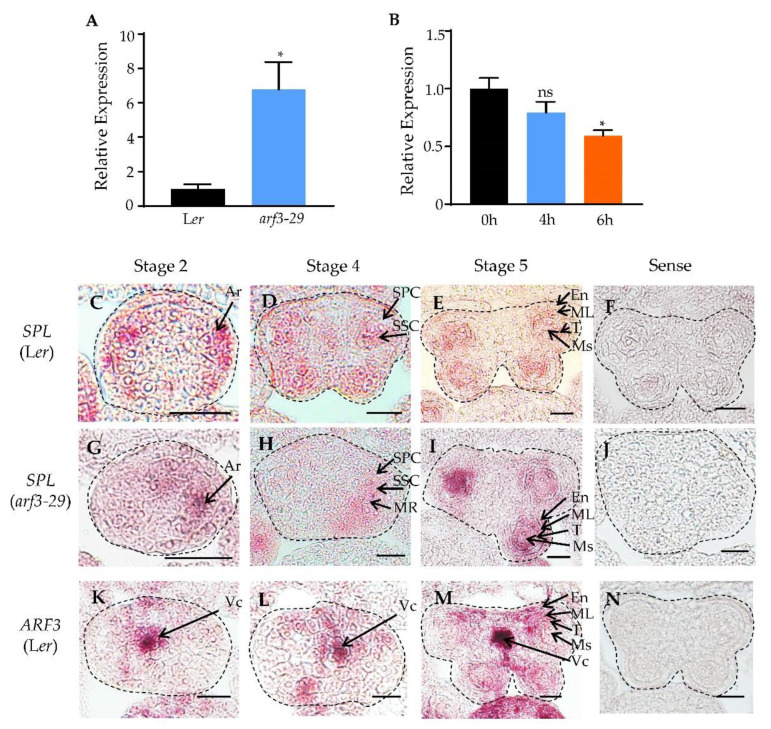
Expression analysis of *SPL* in wild-type and *arf3* mutants. (**A**) Real-time PCR analysis showing the relative expression of *SPL* in *arf3-29*. (**B**) Real-time PCR analysis showing the relative expression of *SPL* in *ARF3::ARF3-GR arf3-29*. The inflorescences were treated with 10 μM DEX once and harvested 0, 4, and 6 h after the treatment for expression analysis. * *p* < 0.05, (Student’s *t*-test), ns, not significant. (**C**–**N**) mRNA in situ hybridization results show the expression of *SPL* in L*er* (**C**–**F**) or *arf3-29* (**G**–**J**) anthers and the expression of *ARF3* in L*er* (**K**–**N**) anthers. The edges of the anthers were marked with black dashed lines. Sections are from anthers of stages 2 to 5 (**C**–**E**,**G**–**J**,**K**–**M**) and controls (**F**,**J**,**N**) using a sense probe on stage 5 anthers. Ar, archesporial cells; SSC, secondary sporogenous cells; SPC, secondary parietal cells; MR, middle region; En, endothecium; ML, middle layer; T, tapetum; Ms, microsporocytes; Vc, vascular. Bars = 20 μm.

**Figure 3 ijms-24-11740-f003:**
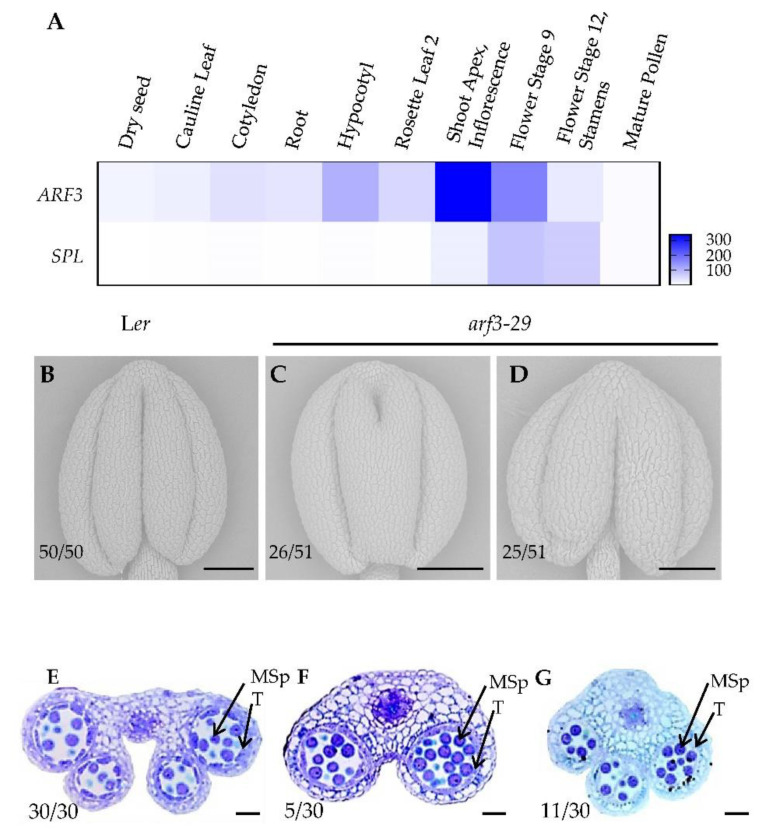
Phenotypic analysis of wild-type and *arf3-29* mutant anthers. (**A**) The heatmap shows the expression levels of *ARF3* and *SPL* in various tissues of *Arabidopsis*. The data come from a public database. (**B**–**D**) Scanning electron microscopy images of L*er* (**B**) and *arf3-29* (**C**,**D**) anthers. Bars = 100 μm. (**E**–**G**) Observation of anther morphology of L*er* (**E**) and *arf3-29* (**F**,**G**) using semi-thin transverse sections. T, tapetum; MSp, microspores. Bars = 50 μm. The number represents the proportion of the corresponding phenotype.

**Figure 4 ijms-24-11740-f004:**
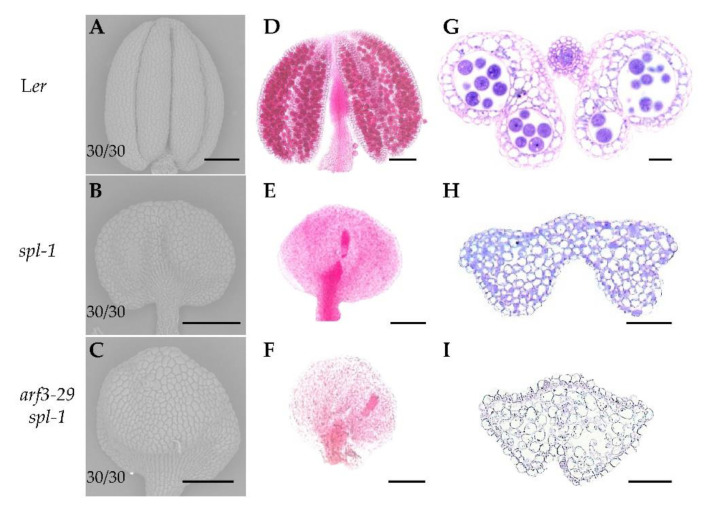
Phenotypic analysis of wild-type, *spl-1* and *arf3-29 spl-1* mutant anthers. (**A**–**C**) Scanning electron microscopy images of L*er* (**A**), *spl-1* (**B**) and *arf3-29 spl-1* (**C**) anthers. The number represents the proportion of the corresponding phenotype. (**D**–**F**) Alexander red staining of pollen grains from L*er* (**D**), *spl-1* (**E**), and *arf3-29 spl-1* (**F**). (**G**–**I**) Morphology of L*er* (**G**), *spl-1* (**H**), and *arf3-29 spl-1* (**I**) anthers using semi-thin transverse sections. Bars = 100 μm (**A**–**F**), 50 μm (**G**–**I**).

**Figure 5 ijms-24-11740-f005:**
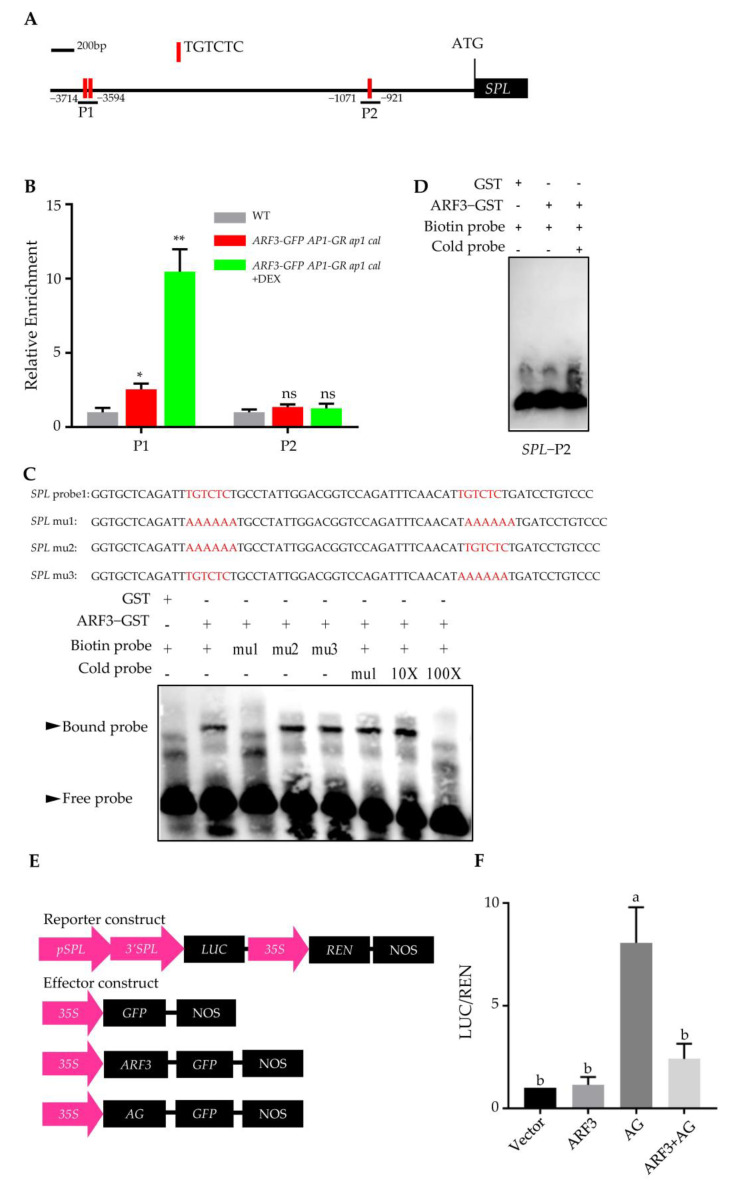
ARF3 directly bound to *SPL* promoter and inhibited AG activation of *SPL* expression. (**A**) Schematic diagram showing the *SPL* promoter region used for ChIP assays in (**B**). (**B**) ARF3 bound analysis by ChIP assays using *ARF3::ARF3-GFP AP1-GR ap1 cal* inflorescences harvested at 5 days after 10 μM DEX treatment. * *p* < 0.05, ** *p* < 0.01 (Student’s *t*-test), ns, not significant. (**C**,**D**) EMSA assays of the binding of ARF3 to the *SPL*-P1 (**C**) and *SPL-P2* (**D**) promoter. (**E**) Structure of the *SPL* promoter and 3′ noncoding region driven dual-LUC reporter gene and two effector genes. For the reporter construct: 35S promoter, *SPL* DNA (3.8 kb promoter and 1 kb 3′ terminal end), REN luciferase (REN), and firefly luciferase (LUC) are indicated. For the effector construct: ARF3 and AG are driven by 35S promoter. (**F**) Transient gene expression assays in *Nicotiana benthamiana*. Data are means ± SD (*n* = 3 biological repeats). Different letters above the bars indicate statistically significant differences between samples.

**Figure 6 ijms-24-11740-f006:**
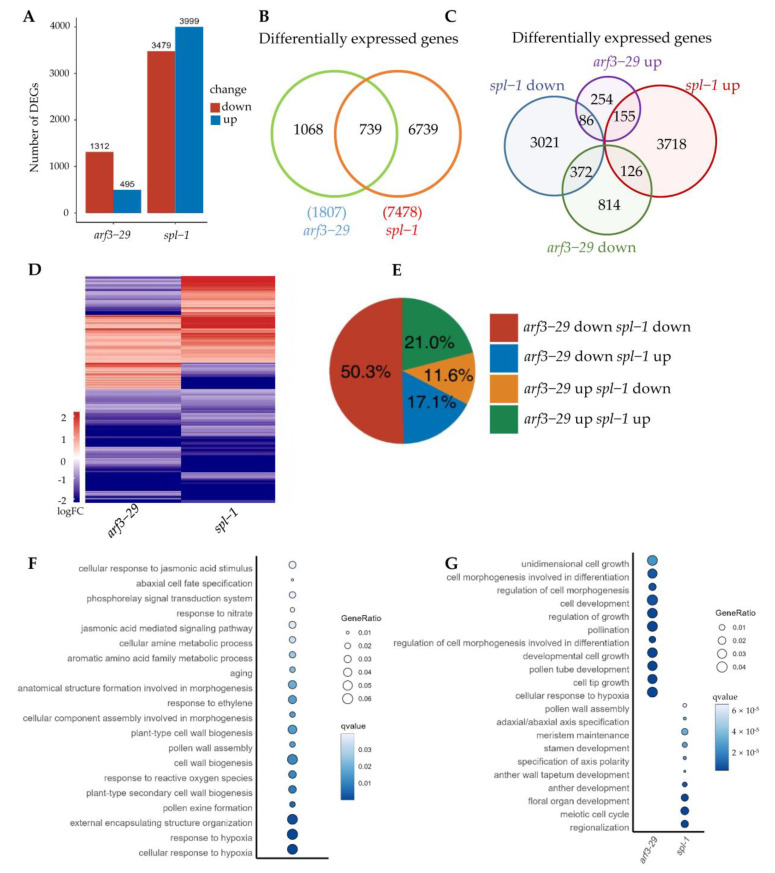
(**A**) Number of up- and down-regulated genes in *arf3-29* and *spl-1* transcriptome data, respectively. (**B**,**C**) Venn diagrams display the common and specific differentially expressed genes (DEGs) between *arf3-29* and *spl-1*. (**D**) The heatmap shows the expression pattern of the 739 common DEGs between *arf3-29* and *spl-1*. (**E**) The 739 common DEGs between *arf3-29* and *spl-1* are classified as four classes. The percentage of each class is indicated. (**F**) Gene Ontology analysis of the common DEGs which regulate in opposite direction (*arf3-29* down *spl-1* up and *arf3-29* up *spl-1* down) in *arf3-29* and *spl-1.* (**G**) Gene Ontology analysis of specific DEGs in *arf-29* or *spl-1*.

**Figure 7 ijms-24-11740-f007:**
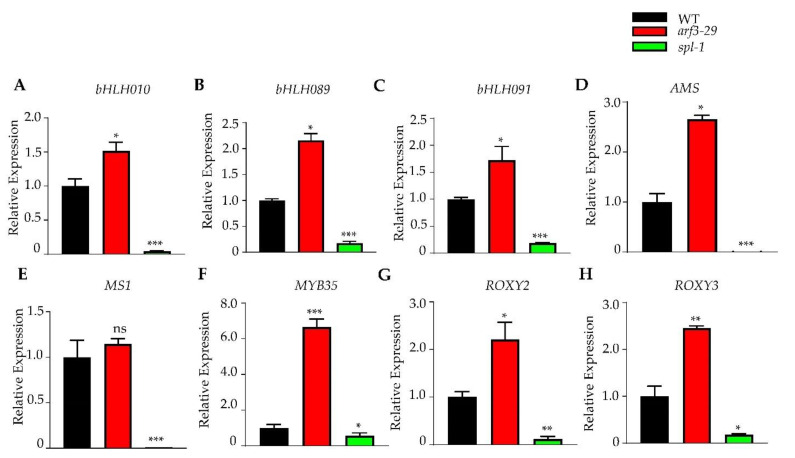
Real-time PCR analysis showing the relative expression of anther regulatory network genes in WT, *arf3-29* and *spl-1* inflorescences. *bHLH010* (**A**), *bHLH089* (**B**), *bHLH091* (**C**), *AMS* (**D**), *MS1* (**E**), *MYB35* (**F**), *ROXY2* (**G**) and *ROXY3* (**H**). * *p* < 0.05, ** *p* < 0.01, *** *p* < 0.001 (Student’s *t*-test), ns, not significant.

## Data Availability

*Arabidopsis* sequence data in this article can be found in the *Arabidopsis* Information Resource (TAIR, http://www.Arabidopsis.org/, accessed on 30 May 2023) under accession nos. *SPL* (AT4G27330), *ARF3* (AT2G33860), *bHLH010* (AT2G31220), *bHLH089* (AT1G06170), *bHLH091* (AT2G31210), *AMS* (AT2G16910), *MS1* (AT5G22260), *MYB35* (AT3G28470), *ROXY2* (AT5G14070), *ROXY3* (AT3G21460). The original transcriptome data from this article were submitted to database under accession number PRJNA975305 (http://www.ncbi.nlm.nih.gov/sra, accessed on 30 May 2023).

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
