# Peer review of "ARF3-Mediated Regulation of SPL in Early Anther Morphogenesis: Maintaining Precise Spatial Distribution and Expression Level"

_ijms, 2023, doi:10.3390/ijms241411740_

Round 1

Reviewer 1 Report

Well-designed experiments that show the regulatory role of ARF3 in the expression of SPL1. Morphological analysis of mutants, in situ hybridization for gene expression localization in anthers, quantitative gene expression, binding of ARF to the SPL promoter and transcriptome analysis provide together compelling evidence that ARF3 regulates the SPL1 gene and that both are required for correct anther and pollen development.

Minor remarks:

-several spelling errors in the text
-Figure 1: I-Q picture are a bit small, difficult to see the different tissues, such as tapetum cells. Maybe an additional row of pictures with specific area at higher magnification would be clarifying.

-Figure 2: the other way around, because of high magnification it is difficult to recognize which part of the anther is presented

typo's such as  page 2 (halfway) "exhibited a defeiciency" deficiency?

page 3 bottom "promounced vacuolization." pronounced?

page 4 5th line from bottom : "To validated.." validate? 

page 6 line 4: "with slight expresses" expression?

Author Response

Review#1

Response: We greatly appreciate the valuable feedback and suggestions provided by the reviewer. We have carefully addressed each of the concerns in the revised manuscript.

  1. Figure 1: I-Q picture are a bit small, difficult to see the different tissues, such as tapetum cells. Maybe an additional row of pictures with specific area at higher magnification would be clarifying.

Response: We thank the reviewer for this suggestion. In response to this comment, we have added two additional rows of pictures at higher magnification. Figure 1R-T provide a closer view of the specific areas highlighted in Figures 1I-K, and Figures 1U-W provide a closer view of the specific areas highlighted in Figures 1L-N. These additional images will enhance the clarity and allow for better visualization of the different tissues, including tapetum cells. It should be noted that in Figures 1L-N, the third phase (Figure 1L) can still form primary parietal cells (PPC) and primary sporogenous cells (PSC) similar to the wild type (Figure 1I), but in the later stages (Figures 1M-N), they fail to differentiate into tapetum (T) and microsporocytes (MSp), resulting in the inability to form normally shaped anther lobes.

  1. Figure 2: the other way around, because of high magnification, it is difficult to recognize which part of the anther is presented.

Response: Thanks for this comment. We have marked the edges of the anthers with black dashed lines. This adjustment will help the reviewer and readers accurately identify the specific regions of the anthers.

  1. Comments on the Quality of English Language

typo's such as page 2 (halfway) "exhibited a defeiciency" deficiency?

page 3 bottom "promounced vacuolization." pronounced?

page 4 5th line from bottom: "To validated.." validate?

page 6 line 4: "with slight expresses" expression?

Response: We thank the reviewer for bring these errors to our attention, and we apologize for any confusion caused. These corrections have been made in the revised manuscript to ensure the accuracy and clarity of the language, including the "promounced" has been revised to “pronounced”; "To validated" has been revised to “to validate”; and "with slight expresses" has been revised to "with slight expression".

Reviewer 2 Report

- The current study focuses on the regulatory mechanisms involved in early anther morphogenesis. This is not a well-studies topic. Novel findings on abnormal SPL mutant phenotypes, spl-4 and spl-5, during anther morphogenesis were reported. The ARF3-SPL module was also found to display an important role in anther polarity development.

- Additional discussion using the reference: 

Li, Q. J., et al. (2007). "The effects of increased expression of an Arabidopsis HD-ZIP gene on leaf morphogenesis and anther dehiscence." Plant Science 173(5): 567-576. 

This reviewer suggests discussing the Arabidopsis semidominant mutant, upcurved leaf1 (ucl1) plants that are male sterile due to anther non-dehiscence at the site where reference 8 is given (Li X, Lian H, Zhao Q, et al. MicroRNA166 monitors SPOROCYTELESS/NOZZLE for building of the anther internal boundary[J]. Plant Physiol, 2019, 181(1): 208-220). The phenotypes of ucl1 were found to be resulted from increased expression of At2g32370, which encodes a member of class IV HD-ZIP protein, HDG3.  This protein was found to be specifically expressed only expressed in anthers. In homozygous ucl1 anthers, the expression levels of three positive regulators of anther dehiscence, MYB26, NST1 and NST2 were down-regulated. These data demonstrate that HDG3 plays a negative role in regulation of anther dehiscence.

- Orthographic errors at a number of places were detected and corrected (colored sites with notes, attached pdf file).

- Ithanticate report of 32% similarity index is too high and it should be reduced, especially at the Materials and Methods section.

Numerous writing errors were corrected (attached pdf file)

Author Response

Review#2

We appreciate the valuable feedback and suggestions provided by the reviewer. In response to the comments, we have made the following revisions in the revised manuscript.

  1. Additional discussion using the reference: Li, Q. J., et al. (2007). "The effects of increased expression of an Arabidopsis HD-ZIP gene on leaf morphogenesis and anther dehiscence." Plant Science 173(5): 567-576.

Response: We thank the reviewer for this suggestion. We have included additional discussion using the reference suggested by the reviewer, Li et al. (2007), to provide further insights into leaf morphogenesis and anther dehiscence.

  1. Ithanticate report of 32% similarity index is too high and it should be reduced, especially at the Materials and Methods section.

Response: Thanks the reviewer for pointing out this. In response to this concern, we have carefully revised the manuscript, especially in the Materials and Methods section, to ensure the similarity index is now lower than 20%.

  1. Comments on the Quality of English Language. Numerous writing errors were corrected (attached pdf file)

Response: We appreciate the reviewer’s thorough review and identification of writing errors.  We have carefully reviewed and corrected all these English mistakes through the manuscript, as highlighted in the pdf files. We apologize for any confusion caused and thank the reviewer for helping us improve the language quality.

  1. Resolution of section 5E needs to be improved.

Response: Thanks. We have improved the resolution of section 5E in the revised manuscript.